# Age-specific SARS-CoV-2 infection fatality rates derived from serological data vary with income and income inequality

Chloe G. Rickards[ID]*, A. Marm Kilpatrick*

Department of Ecology and Evolutionary Biology, University of California Santa Cruz, Santa Cruz, CA, United States of America

* rickards.chloe@gmail.com (CGR); akilpatr@ucsc.edu (AMK)

**Data Availability Statement:** All scripts and data files are available from a public Github repository at https://github.com/ChloeRickards/sars-cov-2-ifr-nyc.

## Abstract

The ongoing COVID-19 pandemic has killed at least 1.1 million people in the United States and over 6.7 million globally. Accurately estimating the age-specific infection fatality rate (IFR) of SARS-CoV-2 for different populations is crucial for assessing and understanding the impact of COVID-19 and for appropriately allocating vaccines and treatments to at-risk groups. We estimated age-specific IFRs of wild-type SARS-CoV-2 using published sero-prevalence, case, and death data from New York City (NYC) from March to May 2020, using a Bayesian framework that accounted for delays between key epidemiological events. IFRs increased 3-4-fold with every 20 years of age, from 0.06% in individuals between 18–45 years old to 4.7% in individuals over 75. We then compared IFRs in NYC to several city- and country-wide estimates including England, Switzerland (Geneva), Sweden (Stockholm), Belgium, Mexico, and Brazil, as well as a global estimate. IFRs in NYC were higher for individuals younger than 65 years old than most other populations, but similar for older individuals. IFRs for age groups less than 65 decreased with income and increased with income inequality measured using the Gini index. These results demonstrate that the age-specific fatality of COVID-19 differs among developed countries and raises questions about factors underlying these differences, including underlying health conditions and healthcare access.

## Introduction

As of January 5, 2023, COVID-19 has killed at least 1.1 million people in the US and over 6.7 million globally [1]. The infection fatality rate (IFR)–the chance of dying after becoming infected–is a crucial metric for understanding the disease severity of SARS-CoV-2 [2–8]. Accurate age-specific IFRs are needed to allocate limited supplies of vaccines, respirators, ICU beds, and anti-viral drugs to minimize mortality from COVID-19 [9,10].

Accurately estimating COVID-19 IFRs requires both quantification of the total number of infections (including undetected asymptomatic and mildly symptomatic infections), and accounting for delays between infection and death [11,12]. First, quantifying the total number of infections is usually done using serosurveys. Accurate IFR estimates require relatively

**Funding:** AMK received funding from the National Science Foundation Division of Environmental Biology, Grants DEB-1717498, and DEB 1911853. CR received funding from the National Science Foundation Graduate Research Fellowship Program. The funders had no role in study design, data collection and analysis, decision to publish, or preparation of the manuscript.

**Competing interests:** The authors have declared that no competing interests exist.

unbiased serosurveys that sample the full population at risk of infection [13]. Second, deaths due to COVID-19 occur within a wide time period after initial infection, (on average, 20.2 days (95% CI 8.0–50.0 [11] after exposure). This time period can be broken down into several smaller delays [11], including the incubation period [14], the delay between symptom onset and case reporting [15], and the delay between case reporting and death [11]. Accurately estimating IFRs requires properly accounting for these delays, as well as the delay between infection and mounting detectable antibodies (seroconversion).

Previous studies of age-specific IFRs for COVID-19 have found a log-linear increase in IFR with age [2–8], except for elevated deaths in very young children. Above 18 years of age, there was a 0.6% increase in IFR with every five years of age [4]. However, there were substantial differences in age-specific IFRs estimates among countries [4,12]. For example, the oldest age groups in Sweden experienced an IFR up to 5 times higher than the oldest groups in Mexico [16,17]. This indicates the need for population-specific age-based IFR estimates when considering the mortality impacts of COVID-19 on a specific community and when assessing potential contributing factors towards increased fatality rates. While the contribution of poverty to COVID-19 mortality has been repeatedly demonstrated [18,19], it is not clear if poverty increases the risk of exposure, the risk of death following infection (the IFR), or both. Surprisingly there have been no age-specific IFR estimates derived solely from a population-based serosurvey of a US population, nor has there been a rigorous comparison of serosurvey-based age-specific IFR estimates between global populations.

We estimated age-specific IFRs using data from a seroprevalence study conducted in New York City shortly after the peak of the spring 2020 epidemic [20], publicly available case and death records [21], and Bayesian inference to account for delays between infection, symptom onset, case ascertainment, death and seroconversion [11]. We then compared these to other IFR estimates that were based on relatively unbiased serosurveys to see if and how age-specific COVID-19 mortality patterns in New York City differed from other populations around the world. Finally, we correlated age-specific IFR estimates from different populations with measures of income and income inequality, which might reflect access to health care.

## Methods

We estimated IFRs using case counts and number of deaths from the New York City Department of Health and Mental Hygiene archive webpage for the dates 3/9/20 to 5/17/20 [21]. To quantify infections, we used data from a serosurvey in New York City conducted over a 10-day period (April 19–28, 2020), two-weeks after the peak in cases in Spring 2020 [20] (Fig 1). Survey participants for the serosurvey were recruited at grocery stores without prior advertisement to reduce bias [20]. The serosurvey was conducted through the entire state of New York, but we focused on New York City for several reasons: New York City had a higher overall seroprevalence (by about 1.5- to 6-fold) and sample size (by about 10-fold) [20], statewide seroprevalence was highly heterogenous [20], and New York City's case and death reporting system is separate from the statewide reporting system [21,22]. Only New York City reported cases and deaths by age, whereas New York State did not, so age-specific IFR estimates were only possible with New York City data [21,22]. The New York City Department of Health and Mental Hygiene reported deaths in two categories: "confirmed" and "combined" (including probable and confirmed deaths). We calculated IFRs for each of these death categories.

To align the seroprevalence survey with reported deaths, we had to address mismatches in the age groups for the two datasets. The New York City serosurvey study reported data in 4 age groups: 18–34, 35–44, 45–54, and 55+ [20]. The New York City Department of Health and Mental Hygiene reported deaths in five age groups: 0–17, 18–44, 45–64, 65–74, and 75+ [21].

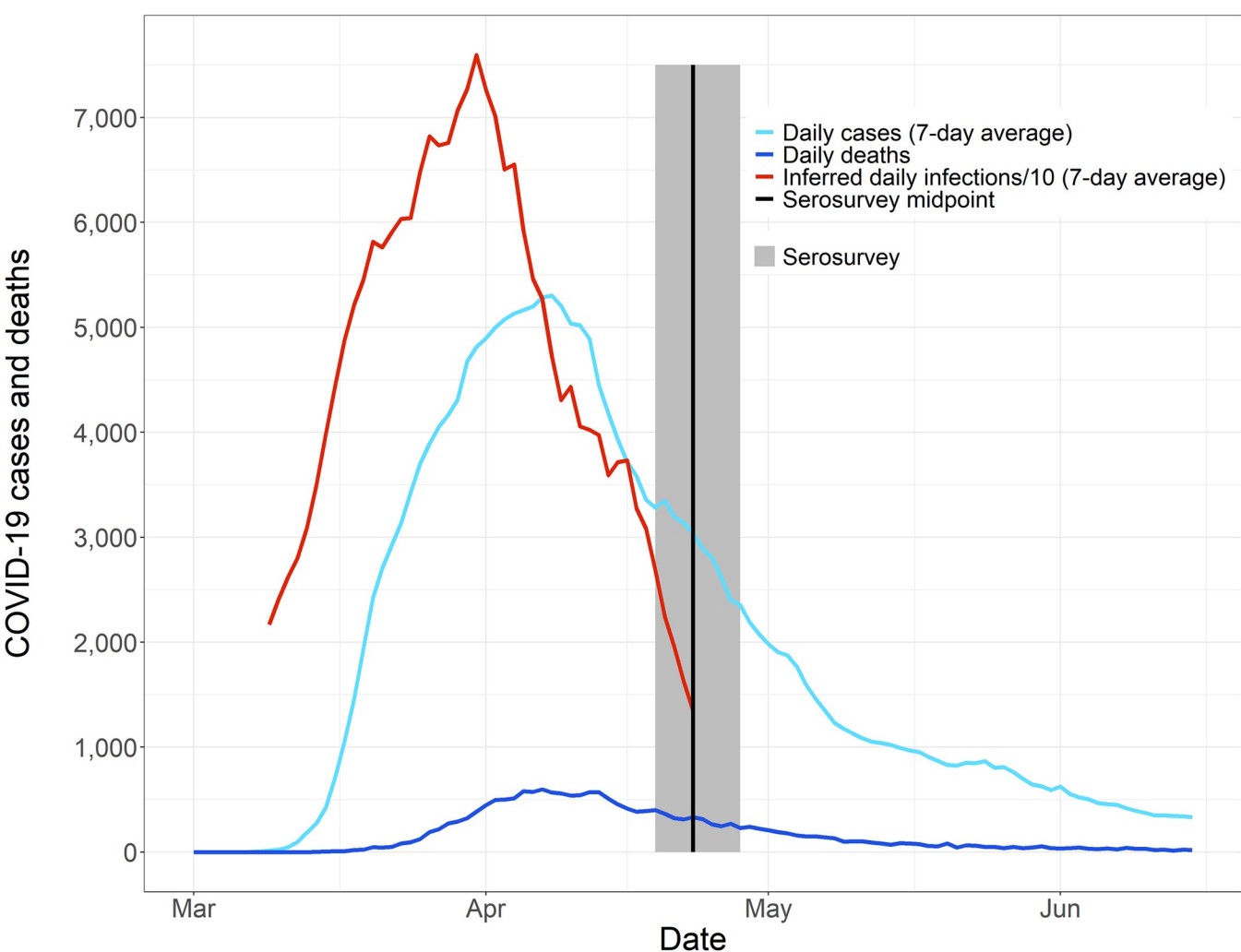

**Fig 1. Delay distributions between infection and key events in the progression of COVID-19.** Key events include symptom onset, case reporting, death, and seroconversion. The distribution of each delay is shown with a violin plot, and boxplots display the median, inter-quartile range (box), and 95% CI (whiskers).

We estimated IFR values for the age classes used for deaths by the New York City Dept. of Health and Mental Hygiene. First, to estimate the seroprevalence for the 0–17 age class, which didn't have a seroprevalence estimate from New York City [20], we estimated the seroprevalence ratio between the 0–17 age class and the 18–44 age class using a serosurvey from Spain, as this survey was one of the few that contained seroprevalence data for 0–17 year olds [23]. This ratio was 0.81, implying an estimated seroprevalence for 0–17 year-old individuals in New York City of 0.81 * 22.3% = 18.1% (8.96–29.8%). Second, we used the single seroprevalence estimate for individuals over the age of 55 [20] for the two oldest age classes used for reported deaths (64–75, and 75+) [21]. Finally, we used a population-weighted average of the 45–54 and 55+ seroprevalences for the 45–64 age class [20,24].

We estimated age-specific IFRs using a previously established Bayesian statistical framework [11], which combines seroprevalence estimates (including uncertainty in the estimates) with time series of cases and deaths [22] and delay distributions between key events [3,6,25–27] (Fig 1; S1 Table). The scripts used to perform this inference are available on Github at https://github.com/ChloeRickards/sars-cov-2-ifr-nyc.

We compared the age-specific IFR estimates for New York City to seven other age-specific IFR estimates that were based on relatively unbiased serosurveys [6,11,16,17,28,29]. We excluded other IFR studies that were not based on population-representative serosurveys, did not properly account for the distribution of delays between infection, seroconversion and death, or did not have reliable case and death data available (see study-specific exclusion criteria in S2 Table). We also compared our estimates to a global age-specific IFR estimate [4]. We used IFR estimates from other studies that included care-home resident deaths because New York City also included care home deaths in their reporting.

## Results

The New York City serosurvey took place in late April, in the latter third of the initial 2020 COVID-19 epidemic, when new cases per day had fallen to approximately half of the peak (Fig 2) [20]. The serosurvey found that 22.7% of the New York City population was seropositive and estimated that approximately 1.5 million infections had occurred [20]. By approximately mid-May, when the last deaths from infections detected in the serosurvey would have occurred, there were nearly 16,000 confirmed COVID-19 deaths (Table 1), and nearly 4,500 probable COVID-19 deaths (S3 Table).

Age-specific IFRs for SARS-CoV-2 in New York City based on confirmed deaths increased logarithmically more than 75-fold from 0.06% in 18–44 to 4.7% in 75+ year-olds (Table 1).

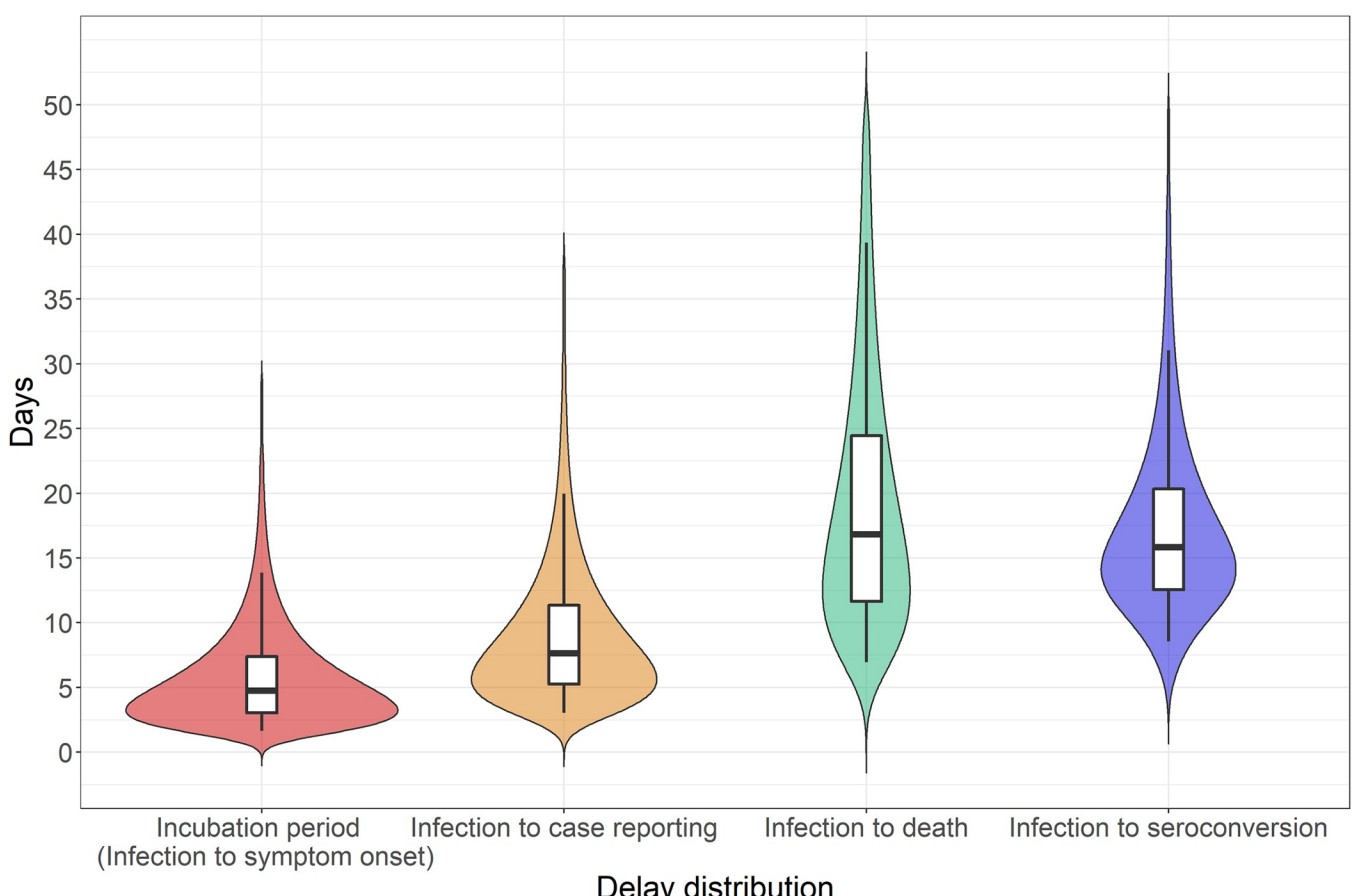

**Fig 2. COVID-19 timeline for New York City in Spring 2020.** The timeline includes confirmed cases, deaths and inferred infections over time, and the duration and midpoint date of the serosurvey. To facilitate display, infections divided by 10 are shown.

**Table 1. Seroprevalence and infection fatality rate (IFR) estimates in New York City for five age classes, using *confirmed* COVID-19 deaths (excluding probable deaths).**

| Age class | Population [24] | Confirmed COVID-19 Deaths, as of May 17, 2020 [21] | Estimated Infection Prevalence [14] (95% CI) | IFR (95% CI) |
|---|---|---|---|---|
| 0–17 | 1,783,174 | 10 | 18.0 (8.96–29.8) | 0.0015 (0.00038–0.0037) |
| 18–44 | 3,493,918 | 625 | 22.4 (19.6–24.9) | 0.063 (0.052–0.076) |
| 45–64 | 2,112,562 | 3556 | 24.1 (21.8–26.4) | 0.52 (0.43–0.61) |
| 65–74 | 689,816 | 3963 | 21.5 (19.6–23.4) | 1.9 (1.6–2.3) |
| 75+ | 551,853 | 7731 | 21.5 (19.6–23.4) | 4.7 (3.9–5.6) |

When including both confirmed and probable COVID-19 deaths, the IFRs were 22–36% higher across adult age classes (S3 Table).

IFRs from New York City for the 18–44 and 45–64 age classes were higher (with non-overlapping 95% confidence intervals) than corresponding IFRs for England [6], Switzerland [11], Belgium [29], Sweden [17], and the global estimate [4] (Figs 3 and S1). In contrast, the IFRs for the oldest two age classes, 65–74 and 75+ were lower (with non-overlapping 95% CIs) than IFRs from England, Switzerland, and the global baseline but were higher than corresponding IFRs from Belgium (Figs 2 and S2). Variation in IFRs among populations decreased with income (Fig 4) and increased with wealth inequality (S2 Fig), as quantified by the Gini index, for younger (18–44, 45–64), but not older (65+), age groups.

## Discussion

We found that IFRs in New York City showed similar log-linear increases with age as many other studies, but there were substantial differences among populations in IFRs, including New York City, for several age groups. The causes for differences in age-specific IFRs among countries are poorly understood [4], but could be due to differences in underlying conditions [19], genetics [30], or poverty and health care access [31], with evidence being limited for two of these three possibilities. First, several underlying health conditions, including obesity, diabetes, immunosuppression, and cancer increase a person's risk of dying from COVID-19 [19], and in New York City, 79% of COVID-19 deaths in 18–44 year-olds, and 85% of deaths in 45–64 year-olds in New York City had pre-existing conditions. A future analysis could examine whether differences among populations in pre-existing conditions underly differences among population in COVID-19 age-specific IFRs. Second, although expression of some genes, like *ace2*, have been linked to both age and increased COVID-19 mortality [32–34], it is unclear if expression of these genes differs among populations.

In contrast, we found correlations among some age-specific IFRs and both income and income inequality, and several other studies have found correlations between COVID-19 mortality and measures of poverty [18,19,31,35]. Studies of poverty and COVID-19 usually haven't been able to separate exposure to SARS-CoV-2 from mortality once infected. As a result, it wasn't previously possible to determine whether poverty led to higher SARS-CoV-2 exposure rates through larger household size, workplace exposure, or other factors, or whether poverty led to higher mortality given infection. Our results suggest that lower income is associated with higher mortality following infection for younger and middle age classes. This may reflect access to health care and medications, or underlying conditions that are exacerbated by poverty [36]. However, a lack of a relationship between income and income inequality and IFRs for the oldest age classes suggests that for these age groups other factors are more important than income.

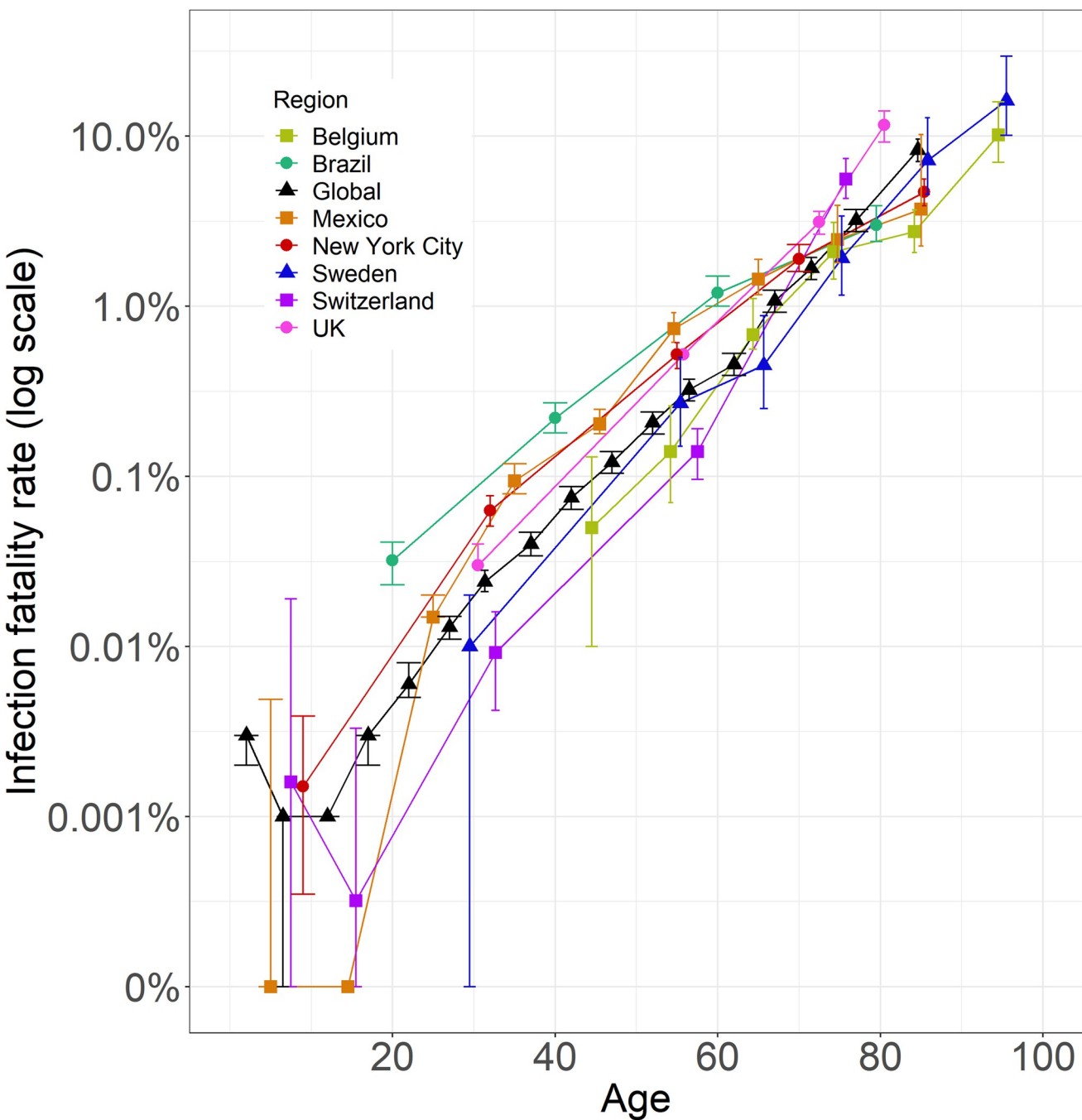

**Fig 3. Age-specific infection fatality ratio (IFR) of COVID-19 on a log-scale (mean ± 95% CI).** Points are plotted at the midpoint of the age-class on the x-axis and are slightly jittered along the x-axis to facilitate presentation. Lines show age-specific IFRs for different populations, using only confirmed COVID-19 deaths where possible. This study is represented as "New York City" in red. IFR means and confidence intervals that estimate a value of 0 are represented as 0.001.

One shortcoming of our study is that age and death data were only available in relatively wide age categories. For example, the oldest age class for the seroprevalence study was 55 + which combined parts of three age categories of the death data (45–64, 65–74, and 75+) which sometimes differ in SARS-CoV-2 infection [6]. Similarly, COVID-19 deaths for 18–44

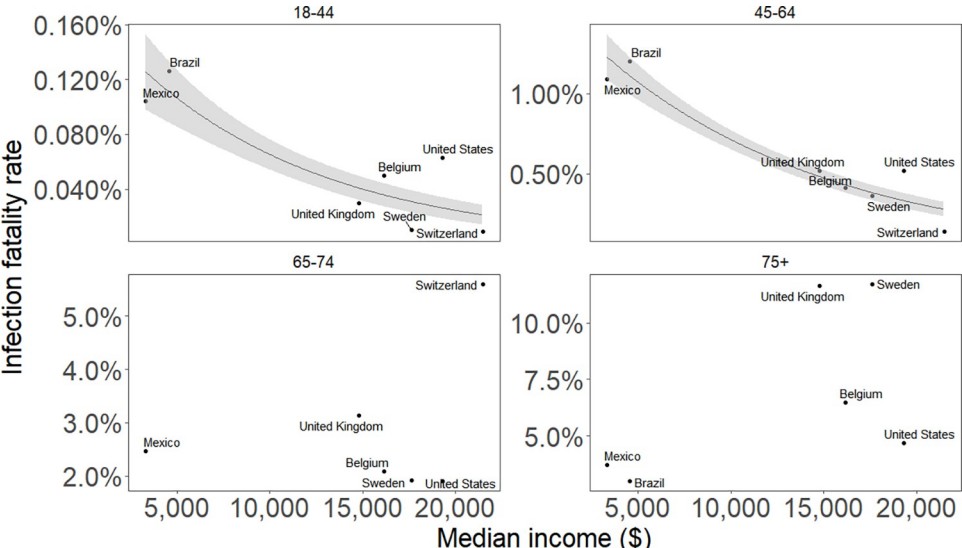

**Fig 4. Infection fatality ratio (IFR) of COVID-19 plotted against median income for that country.** Data points shown for age groups (a) 18–44, (b) 45–64, (c) 65–74, and (d) 75+. Fitted lines show significant relationships where they exist. There were only three IFR estimates for the 0–17 age category so this age category was omitted from the plot.

year olds were combined into one group, despite variation in seroprevalence within this age group [6]. Reporting seroprevalence and deaths for finer age classes would enable more accurate estimates and a better understanding of the fatality of COVID-19 across age groups.

The COVID-19 pandemic has reshaped communities worldwide. We found substantial variation in age-specific infection fatality rates for COVID-19, both between richer and poorer countries and even within wealthier countries. This variation can have substantial consequences. For example, in addition to variation in mortality, COVID-19 deaths have led to millions of cases of caregiver death and orphanhood [37,38]. Notably, age-specific COVID-19 infection fatality rates were correlated with both income and income inequality, suggesting that the disproportionate effects of COVID-19 on poorer communities arose not only from higher exposure rates, but also from higher mortality following infection. A key goal in preparing for the next pandemic is reducing inequalities that contribute to these higher infection fatality rates.

## Supporting information

**S1 Fig. Age-specific infection fatality ratio (IFR) of COVID-19 (mean ± 95% CI) on an untransformed scale.** Points are plotted at the midpoint of the age class on the x-axis, and slightly jittered along the x-axis facilitate presentation. Lines show age-specific IFRs for different populations, using confirmed-only COVID-19 deaths where possible. This study is represented in red, as "New York City". IFR means and confidence intervals that estimate a value of 0 are represented as 0.001. The upper bound on the 90+ class for Sweden extends to 29.5% but is represented here as 20%.
(PDF)

**S2 Fig. Infection fatality ratio (IFR) of COVID-19 plotted against the Gini index [41].** Higher values of the Gini index represent higher levels of inequality. Fitted lines show significant relationships. Age classes include (a) 18–44, (b) 45–64, (c) 65–74, and (d) 75+. There were

only three IFR estimates for the 0–17 age category so this age category was omitted from the plot.
(PDF)

**S1 Table. Delay distributions and sources.** The delays from symptom onset to case reporting [27] and from case reporting to death [28] are specific to New York City. See Perez-Saez et al. [11] for additional methods on delay distributions.
(PDF)

**S2 Table. Exclusion criteria for global age-specific IFR comparisons.** Serosurveys based on blood donors were not considered.
(PDF)

**S3 Table. Seroprevalence and infection fatality rate (IFR) estimates in New York City for five age classes, using confirmed and probable COVID-19 deaths.** The last column notes the percent increase in the IFR comparing confirmed deaths to confirmed and probable deaths.
(PDF)

## Acknowledgments

We thank the authors of Reference [11] for making their R code available which facilitated the analyses presented here. We thank members of the Kilpatrick lab for feedback.

## Author Contributions

**Conceptualization:** Chloe G. Rickards, A. Marm Kilpatrick.

**Data curation:** Chloe G. Rickards.

**Formal analysis:** Chloe G. Rickards.

**Funding acquisition:** Chloe G. Rickards.

**Investigation:** Chloe G. Rickards, A. Marm Kilpatrick.

**Methodology:** Chloe G. Rickards, A. Marm Kilpatrick.

**Project administration:** A. Marm Kilpatrick.

**Resources:** A. Marm Kilpatrick.

**Software:** Chloe G. Rickards.

**Supervision:** A. Marm Kilpatrick.

**Validation:** Chloe G. Rickards, A. Marm Kilpatrick.

**Visualization:** Chloe G. Rickards, A. Marm Kilpatrick.

**Writing – original draft:** Chloe G. Rickards.

**Writing – review & editing:** Chloe G. Rickards, A. Marm Kilpatrick.

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
