## [Decision Letter · Decision Letter 0]

20 Feb 2023

PONE-D-23-00943Age-specific SARS-CoV-2 infection fatality rates derived from serological data vary with income and income inequalityPLOS ONE

Dear Dr. Rickards,

Thank you for submitting your manuscript to PLOS ONE. After careful consideration, we feel that it has merit but does not fully meet PLOS ONE’s publication criteria as it currently stands. Therefore, we invite you to submit a revised version of the manuscript that addresses the points raised during the review process.

We look forward to receiving your revised manuscript.

Kind regards,

Junyuan Yang

Academic Editor

PLOS ONE

Journal Requirements:

" ext-link-type="uri" xlink:type="simple">https://journals.plos.org/plosone/s/file?id=ba62/PLOSOne_formatting_sample_title_authors_affiliations.pdf"

2. Thank you for including your ethics statement:  "N/A".   

a. For studies reporting research involving human participants, PLOS ONE requires authors to confirm that this specific study was reviewed and approved by an institutional review board (ethics committee) before the study began. Please provide the specific name of the ethics committee/IRB that approved your study, or explain why you did not seek approval in this case.

b. Please provide additional details regarding participant consent. In the ethics statement in the Methods and online submission information, please ensure that you have specified (1) whether consent was informed and (2) what type you obtained (for instance, written or verbal, and if verbal, how it was documented and witnessed). If your study included minors, state whether you obtained consent from parents or guardians. If the need for consent was waived by the ethics committee, please include this information.

Reviewers' comments:

Reviewer's Responses to Questions

**Comments to the Author**

1. Is the manuscript technically sound, and do the data support the conclusions?

Reviewer #1: Yes

Reviewer #2: Partly

2. Has the statistical analysis been performed appropriately and rigorously? 

Reviewer #1: Yes

Reviewer #2: N/A

3. Have the authors made all data underlying the findings in their manuscript fully available?

Reviewer #1: Yes

Reviewer #2: No

4. Is the manuscript presented in an intelligible fashion and written in standard English?

Reviewer #1: Yes

Reviewer #2: Yes

5. Review Comments to the Author

Reviewer #1: This paper estimated age-specific IFRs using data from a seroprevalence study conducted in New York City shortly after the peak of the spring 2020 epidemic. Then compared these to other IFR estimates that were based on relatively unbiased serosurveys to see if and how age-specific COVID-19 mortality patterns in New York City differed from other populations around the world. Finally, it correlated age-specific IFR estimates from different populations with measures of income and income inequality, which might reflect access to health care. This is helpful to provide some suggestions about the problem of the allocation of medical resources. The author is expected to make subsequent modifications according to the requirements and specifications of the journal.

Reviewer #2: The infection fatality and the age of the patients have been described. However, the authors need to explain the infection fatality rate and its relation to the income inequality. The authors must segregate the countries referenced in to /based on income differences. The authors, although have provided a graphical representation, it is not clear about the income difference that was one of the aspects in the hypotheses/title.

6. PLOS authors have the option to publish the peer review history of their article (what does this mean?). If published, this will include your full peer review and any attached files.

Reviewer #1: No

Reviewer #2: **Yes: **Dr. Venkataramana Kandi

---

## [Author Response · Author response to Decision Letter 0]

17 Mar 2023

Ethics Statement:

This study used previously published data collected under a New York State Department of Health IRB. No original human research data were collected for this study and all death data were anonymized.

Editorial comments:

 The revised manuscript is formatted in accordance with PLOS ONE’s style requirements.

2. Ethics statement:

 In this manuscript, we used published data which was collected under an IRB. This manuscript does not contain any original human research data. We have updated our Methods and Ethics Statement to clarify this.

Reviewer comments:

Reviewer #1: This paper estimated age-specific IFRs using data from a seroprevalence study conducted in New York City shortly after the peak of the spring 2020 epidemic. Then compared these to other IFR estimates that were based on relatively unbiased serosurveys to see if and how age-specific COVID-19 mortality patterns in New York City differed from other populations around the world. Finally, it correlated age-specific IFR estimates from different populations with measures of income and income inequality, which might reflect access to health care. This is helpful to provide some suggestions about the problem of the allocation of medical resources. The author is expected to make subsequent modifications according to the requirements and specifications of the journal.

 We appreciate the positive comments from the reviewer and have revised the ms to meet the journal style requirements.

Reviewer #2: The infection fatality and the age of the patients have been described. However, the authors need to explain the infection fatality rate and its relation to the income inequality. The authors must segregate the countries referenced in to /based on income differences. The authors, although have provided a graphical representation, it is not clear about the income difference that was one of the aspects in the hypotheses/title.

 We have revised the text to clarify the relationship between income and IFRs.

---

## [Decision Letter · Decision Letter 1]

27 Apr 2023

Age-specific SARS-CoV-2 infection fatality rates derived from serological data vary with income and income inequality

PONE-D-23-00943R1

Dear Dr. Richard,

We’re pleased to inform you that your manuscript has been judged scientifically suitable for publication and will be formally accepted for publication once it meets all outstanding technical requirements.

Kind regards,

Junyuan Yang

Academic Editor

PLOS ONE

Additional Editor Comments (optional):

Reviewers' comments:

Reviewer's Responses to Questions

**Comments to the Author**

1. If the authors have adequately addressed your comments raised in a previous round of review and you feel that this manuscript is now acceptable for publication, you may indicate that here to bypass the “Comments to the Author” section, enter your conflict of interest statement in the “Confidential to Editor” section, and submit your "Accept" recommendation.

Reviewer #1: (No Response)

2. Is the manuscript technically sound, and do the data support the conclusions?

Reviewer #1: Partly

3. Has the statistical analysis been performed appropriately and rigorously? 

Reviewer #1: No

4. Have the authors made all data underlying the findings in their manuscript fully available?

Reviewer #1: Yes

5. Is the manuscript presented in an intelligible fashion and written in standard English?

Reviewer #1: Yes

6. Review Comments to the Author

Reviewer #1: 1.Please correct the word "an" as "a" in line 96.

2.There is a problem: Is there less supporting data showing a correlation between income inequality and IFRs? The Figure 4 in this paper only tells us that death rates in some age groups(18-44 and 45-64) may be related to income inequality.

7. PLOS authors have the option to publish the peer review history of their article (what does this mean?). If published, this will include your full peer review and any attached files.

Reviewer #1: No

---

## [Editor Report · Acceptance letter]

4 May 2023

PONE-D-23-00943R1 

Age-specific SARS-CoV-2 infection fatality rates derived from serological data vary with income and income inequality 

Dear Dr. Rickards:

I'm pleased to inform you that your manuscript has been deemed suitable for publication in PLOS ONE. Congratulations! Your manuscript is now with our production department. 

Kind regards, 

on behalf of

Dr. Junyuan Yang 

Academic Editor

PLOS ONE